# Global Position Analysis during Official Elite Female Beach Volleyball Competition: A Pilot Study

Paulo Vicente João [1,2,*], Alexandre Medeiros [3,4], Henrique Ortigão [5], Mike Lee [5] and Maria Paula Mota [1,2]

1 Research Center in Sports Science, Health and Human Development (CIDESD), 5000-801 Vila Real, Portugal; mpmota@utad.pt
2 Department of Sport Science, Exercise and Health, University of Trás-os-Montes and Alto Douro (UTAD), 5000-801 Vila Real, Portugal
3 Institute of Physical Education and Sport, Federal University of Ceara, Fortaleza 60455-760, Brazil; alexandreararaipe@hotmail.com
4 Master Program in Physiotherapy and Functioning, Federal University of Ceara, Fortaleza 60455-760, Brazil
5 Catapult Sports, Leeds LS2 7JU, UK; Henrique.ortigao@catapultsports.com (H.O.); mike.lee@catapultsports.com (M.L.)
* Correspondence: pvicente@utad.pt; Tel.: +351-965884853

**Abstract:** The aim of this study was to quantify the physical demands of female beach volleyball competition with reference to player position, set, and match outcome. Twelve professional players were equipped with a 10 Hz GPS device (Minimax S4, Catapult Sports, Australia). Data collection occurred over 30 official matches, with a total of 50 sets. GPS output variables were related to position (e.g., Defenders and Blockers). Differences between players' positions were found in Peak Player Load, the distance covered at different intensities, and acceleration and deceleration. Variations during the match were more pronounced for Defenders than for Blockers, with the former increasing the intensity of acceleration and deceleration, and decreasing the velocity of displacements and lower jumps. For Blockers, main variations occurred between the first and second set, with a reduction in velocity displacements and an increase in the intensity of jumps. Defender variables that contributed to victory were high deceleration, velocity, acceleration, and Peak Player Load. The characteristics of Blockers that contributed to victory were maximum velocity and high jumps. Female beach volleyball players seem to have different physiological requirements according to their position. The analysis of these variations throughout the game suggests that a specific player's position output may be determined by proper and/or opponent tactical schemes.

**Keywords:** match load; GPS; players' position; set; outcome; sport physiology

## 1. Introduction

The popularity of beach volleyball (BV) has increased. As a result, researchers have been conducting research on the factors that affect athletes' performance and match outcomes. Previously, some important studies have contributed to the increase in the knowledge of the physiological response [1,2], technical and tactical aspects [3], training load [4–6], and match analysis of BV [7–9]. Most of the findings revealed that BV is performed intermittently, with brief bouts of high-intensity exercise interspersed by low-intensity periods [10]. Understanding the contributions of power output, environment, skill, and subjective scoring to the variability of athletic performance should us help to identify and define strategies for performance enhancement [11]. A BV competition involves several matches in the same day and the following days (varying from four to eight matches), which increases the need to characterize various efforts during each game, as well as the ability to recover from fatigue between games [12,13]. According to Magalhães et al. [10], a BV match induces a temporary reduction in lower limb strength and sprinting time. A decrease in knee extensor and flexor muscles (maximal isometric voluntary contraction) at 0 h (~19% and

17%, respective-ly) was also reported, but after 3 h these values had already returned to the baseline. Although this information is important as several matches take place in the same day, it appears essential to measure the density and intensity of explosive movements during the match and analyze how their performance varies alongside it. The Global Positioning System (GPS), and associated technology (GPS technology), can be useful for the quantification and characterization of the type, duration, and frequency of discrete movements, resulting in a dynamic profile of BV match play [14,15]. Moreover, GPS data may help to detect fatigue during matches, identify periods of the most intense play, define different activity profiles based on position (Blocker or Defender) that are associated with game-specific tasks, and tactical or strategic information [14]. However, only a few studies have employed GPS technology to characterize the movement pat-terns of elite BV match play [9,16,17]. The pilot study of João et al. [17], revealed an average distance of 1800 m per match (three sets), performed intermittently, at high intensity (effort was above 85% of max-imal heart rate for most of the duration). These results were confirmed by Nunes et al. [9], who followed an Olympic female BV team over 99 national and international games and found that these players per-formed most of the total distance covered during match play (~85%) at low-speed (i.e., walking and jogging) and used low acceleration and deceleration movements ($<2$ m.s$^{-2}$). However, almost more than 55% of the time was spent with a heart rate that was over 80% of the maximal heart rate. One study [18] found significant differ-ences between Blockers and Defenders in terms of the number of jumps performed, despite no reference to the height being made. More recently, Bellinger et al. [16], highlighted the differences in the external load of female BV players according to their age, score margin, and set-to-set variations. These researchers demonstrated that adult BV players covered a greater relative distance in zones of higher intensity compared to youth athletes. Moreover, sets that were decided by smaller score margins (<6-point score differential) demanded a greater relative distance, peak speed, greater mean acceleration, and deceleration output, as well as a greater relative distance in speed zones one and three compared to sets decided by larger score margins (>5-point score differential). These studies analyzed important variables concerned with the characterization of female BV match play. However, only João et al. [17], analyzed differences between players according to their position, and none of the studies used GPS technology to measure jump volume during matches, which can be a determinant of success in major volleyball actions such as block, spike, and service. The monitoring of physical demand can provide important information that will allow staff to use more effective training prescriptions to maximize athletes' performance, encourage recovery, and prevent in-jury. Therefore, the aim of this pilot study was to quantify, using GPS technology, external workload (e.g., Player Load, distance covered, acceleration and deceleration, maximal speed, and jumps) to predict the final scores of the Women BV Por-tuguese Championship according to player position (Blocker vs. Defender). The specific aims of the study were: (a) to characterize athletes' profiles by position based on external workload variables, (b) to analyze external workload variables over the three sets with reference to player position, and (c) to analyze the external workload variables that most likely lead to victory or defeat for each player position.

## 2. Materials and Methods

Six Defenders (age: 26.3 ± 5.8 years; height: 175.3 ± 5.5 cm; weight: 64.6 ± 4.0 kg; training experience: 9.6 ± 4.9 years) and six Blockers (age: 29.0 ± 7.7 years; height: 178.0 ± 5.2 cm; weight: 66.1 ± 7.7 kg; training experience: 12.1 ± 6.1 years) were studied. The top six female BV teams from the Portuguese National Beach Volleyball Championship (composed of 38 teams) took part. All the players competed at professional volleyball championships and had at least three years' experience of playing BV as professionals (10.9 ± 5.5 years). All subjects were healthy, with no recent injuries, and had a physical examination and were cleared of any medical disorders that might limit their participation. Written informed consent, according to the Declaration of Helsinki, was obtained from

players before the investigation, and the University of Trás-os-Montes and the Alto Douro Ethics Committee approved the study (REF.: Doc25-CE-UTAD-2016).

Players were categorized by position (Blockers n = 6, Defenders n = 6), and data collection occurred over 30 official matches, with a total of 50 sets (16 first and second sets, 6 third sets, and 8 byes) during a tournament of the national championship. All the players were equipped with a commercially available 10 Hz GPS device containing an inertial measurement unit (Minimax S4, Catapult Sports, Melbourne, Australia), positioned in a harness between the C7 and T2 vertebrae. Each GPS was activated at least 10 min before matches began. Before testing, the GPS was activated and left for 10 min. This allowed the unit to download ephemeris data from satellites used to calculate location and distance, and data collection was monitored in real time. For data analysis, only data recorded during the match were considered (periods during time outs and intervals were not considered). The details of each positional variable can be seen in Table 1. Variables that express different levels of intensity (for example, displacement can be covered walking, jogging, or quick running) were expressed as the percentage of the total.

**Table 1.** Descriptions of positional variables.

| Variables | | Unit | Description |
| --- | --- | --- | --- |
| Distance covered (m) | Total | m | Total distance covered in meters |
| Relative distance covered (m/min) | Total | m/min | Total distance covered in meters per minute |
| Velocity Low | Walking | (0–3.9 km/h) m/min | Total distance covered between 0 and 3.9 km/h/min$^{-1}$ |
| Velocity Medium | Jogging | (4.0–6.9 km/h) m/min | Total distance covered between 4.0 and 6.9 km/h/min$^{-1}$ |
| Velocity High | Quick Running | (7.0–12.9 km/h) m/min | Total distance covered between 7.0 and 12.9 km/h/min$^{-1}$ |
| Maximum velocity (m.s$^{-2}$) | Total | SpeedAVG | Average max speed |
| Accelerations | L/M/H | ACC (>2 m/s$^2$) n/min | Total positive speed changes per minute (Low, Medium, High) |
| Decelerations | L/M/H | DEC (> 2 m/s$^2$) n/min | Total negative speed changes per minute (Low, Medium, High) |
| Jumps | L/M/H | JUM/n/min 400-ms flight time | Total number of jumps recorded per minute (Low, Medium, High) |
| Player load (a.u) | Total | PL/a.u./min | Accumulated accelerometer load in the three axes of movement |
| Peak Player load (a.u) | Total | PL/a.u./min | Accumulated Máxim accelerometer load in the three axes of movement |

The normality of the distribution of the variables was checked using the Shapiro–Wilk test. Values are reported as mean ± SD. The Mann–Whitney test was used to compare each variable between positions (Defender vs. Blocker). The comparison be-tween sets was made between the first and second set, and between the second and fourth set, using the Wilcoxon test. Discriminant analysis with a 95% confidence in-terval (CI) was used to identify players' position variables; it was able to explain the differences between winners and losers using a coefficient greater than |SC| ≥ 30 (Tabachnick and Fidell, 2013). Effect size (ES) was calculated using Cohen's d [19]. The analyses were performed using the SPSS 27.0 statistical software package (SPSS Inc., Chicago, IL, USA), with a significance level set at 5%.

## 3. Results

Table 2 presents the average values (± SD) and comparison per set of external load variables of female defending and blocking BV players. Differences between players' positions were found in the Peak Player Load, with Defenders presenting higher values than Blockers. Blockers performed a higher distance walking, but were lower in jogging and quick running intensity than Defenders. The acceleration of Blockers at medium intensity was higher than that of Defenders. Regarding deceleration, Blockers presented higher values at a lower intensity, while Defenders showed higher values at medium- and higher-intensity zones. In relation to Effect Size (ES) values, the comparisons were small in almost all variables, which mean that the difference is negligible, even if it is statistically significant. According to the ES values, differences between player's position had medium relevance for the velocity.

**Table 2.** Descriptive statistics ($x \pm$ SD) and comparison of GPS variables according to the specific position (Defender and Blocker) in female beach volleyball.

| Variables | Position | $x \pm$ SD | IC 95% | ES | Z | $p$ |
|---|---|---|---|---|---|---|
| Distance Covered (m) | Defender | $543.1 \pm 195.6$ | 529.3–556.7 | $-0.02$ | $-1.470$ | 0.142 |
| | Blocker | $539.2 \pm 201.5$ | 524.9–553.5 | | | |
| Player Load (AU) | Defender | $75.5 \pm 28.8$ | 73.5–77.6 | 0.03 | $-0.861$ | 0.389 |
| | Blocker | $76.3 \pm 29.7$ | 74.2–78.4 | | | |
| Peak Player Load (min) | Defender | $6.6 \pm 3.1$ | 6.4–6.8 | $-0.20$ | $-6.130$ | 0.000 * |
| | Blocker | $5.9 \pm 3.3$ | 5.6–6.1 | | | |
| Meterage per Min (m.min$^{-1}$) | Defender | $32.9 \pm 6.6$ | 32.3–33.3 | $-0.06$ | $-0.998$ | 0.318 |
| | Blocker | $32.5 \pm 6.4$ | 32.0–33.0 | | | |
| Maximum Velocity (m.s$^{-2}$) | Defender | $10.4 \pm 2.5$ | 10.2–10.5 | 0.04 | $-0.033$ | 0.974 |
| | Blocker | $10.5 \pm 2.5$ | 10.3–10.6 | | | |
| Velocity Low: Walking (%) | Defender | $85.4 \pm 21.6$ | 83.8–86.9 | 0.59 | $-6.879$ | 0.000 * |
| | Blocker | $95.4 \pm 4.1$ | 95.1–95.7 | | | |
| Velocity Medium: Jogging (%) | Defender | $13.1 \pm 18.7$ | 11.8–14.4 | $-0.59$ | $-6.913$ | 0.000 * |
| | Blocker | $4.4 \pm 3.8$ | 4.1–4.7 | | | |
| Velocity High: Quick running (%) | Defender | $1.4 \pm 3.1$ | 1.2–1.6 | $-0.54$ | $-10.995$ | 0.000 * |
| | Blocker | $0.1 \pm 0.5$ | 0.04–0.12 | | | |
| Acceleration Low (%) | Defender | $61.7 \pm 17.1$ | 60.5–63.0 | $-0.01$ | $-1.058$ | 0.290 |
| | Blocker | $61.6 \pm 16.0$ | 60.4–62.7 | | | |
| Acceleration Medium (%) | Defender | $21.3 \pm 10.4$ | 20.5–22.0 | 0.11 | $-2.101$ | 0.036 * |
| | Blocker | $22.5 \pm 10.5$ | 21.8–23.3 | | | |
| Acceleration High (%) | Defender | $16.9 \pm 13.1$ | 16.0–17.8 | $-0.08$ | $-1.868$ | 0.062 |
| | Blocker | $15.8 \pm 12.5$ | 14.9–16.6 | | | |
| Deceleration Low (%) | Defender | $57.9 \pm 15.0$ | 56.9–59.0 | 0.29 | $-6.736$ | 0.000 * |
| | Blocker | $62.4 \pm 13.6$ | 61.4–63.3 | | | |
| Deceleration Medium (%) | Defender | $27.7 \pm 12.1$ | 26.9–28.6 | $-0.21$ | $-4.801$ | 0.000 * |
| | Blocker | $25.2 \pm 9.8$ | 24.6–26.0 | | | |
| Deceleration High (%) | Defender | $14.2 \pm 13.3$ | 13.2–15.1 | $-0.15$ | $-2.048$ | 0.041 * |
| | Blocker | $12.2 \pm 11.3$ | 11.4–13.0 | | | |
| Jump Low (<20 cm) (%) | Defender | $17.0 \pm 15.6$ | 15.9–18.1 | $-0.02$ | $-1.021$ | 0.307 |
| | Blocker | $16.6 \pm 14.4$ | 15.6–17.7 | | | |
| Jump Medium (20–40 cm) (%) | Defender | $53.2 \pm 22.1$ | 51.7–54.8 | $-0.23$ | $-4.847$ | 0.000 * |
| | Blocker | $47.7 \pm 22.0$ | 46.1–49.2 | | | |
| Jump High (>40 cm) (%) | Defender | $29.7 \pm 21.6$ | 28.1–31.2 | 0.24 | $-4.602$ | 0.000 * |
| | Blocker | $35.6 \pm 24.5$ | 33.8–37.3 | | | |

* $p \leq 0.05$.

Table 3 shows the external load during each set by position. The total distance covered by Defenders and Blockers decreased from the second to the third set ($p = 0.000$ for both comparisons). The same variation was observed for the Player Load ($p = 0.000$ for both comparisons). A significant decrease was found in the Peak Player Load between the second and the third sets for Defenders ($p = 0.045$). Both players significantly decreased the meterage per minute between the first and the second set ($p = 0.001$ for Defenders and $p = 0.000$ for Blockers). Defenders decreased Maximal Velocity in the second set compared to the first ($p = 0.018$).

**Table 3.** Descriptive statistics ($x \pm$ SD) and comparison of GPS variables according to player position and set in female beach volleyball.

| Variables | Position | Set 1 ($x \pm$ SD) | Set 2 ($x \pm$ SD) | Set 3 ($x \pm$ SD) |
|---|---|---|---|---|
| Distance covered (m) | Defender | 556.1 ± 201.2 | 569.6 ± 161.7 | 442.7 ± 231.8 [b] |
| | Blocker | 559.4 ± 206.5 | 546.8 ± 160.3 | 460.5 ± 254.4 [b] |
| Player Load (AU) | Defender | 75.4 ± 26.6 | 80.7 ± 25.7 | 62.1 ± 36.3 [b] |
| | Blocker | 77.8 ± 28.9 | 78.5 ± 25.8 | 65.0 ± 39.4 [b] |
| Peak Player Load (AU) | Defender | 6.4 ± 2.3 | 6.8 ± 3.1 | 6.3 ± 3.0 [b] |
| | Blocker | 5.6 ± 1.9 | 6.3 ± 3.6 | 6.4 ± 2.6 |
| Meterage per Min (m/min) | Defender | 33.6 ± 5.9 | 32.0 ± 5.6 [a] | 33.3 ± 9.9 |
| | Blocker | 33.7 ± 6.6 | 31.1 ± 5.2 [a] | 33.7 ± 8.5 |
| Maximum Velocity (m.s$^{-2}$) | Defender | 10.6 ± 2.3 | 10.2 ± 2.4 [a] | 10.2 ± 3.2 [b] |
| | Blocker | 10.4 ± 2.1 | 10.4 ± 2.4 | 10.8 ± 3.4 |
| Velocity Low: Walking (%) | Defender | 84.0 ± 23.0 | 85.3 ± 21.6 | 88.5 ± 17.9 [b] |
| | Blocker | 95.0 ± 5.1 | 96.0 ± 2.9 [a] | 95.1 ± 4.3 |
| Velocity Medium: Jogging (%) | Defender | 14.2 ± 19.7 | 13.2 ± 18.6 | 10.6 ± 16.5 [b] |
| | Blocker | 4.8 ± 4.5 | 3.9 ± 2.9 [a] | 4.7 ± 4.1 |
| Velocity High: Quick running (%) | Defender | 1.7 ± 3.5 | 1.5 ± 3.2 | 0.7 ± 1.5 |
| | Blocker | 0.1 ± 0.7 | 0.04 ± 0.2 | 0.2 ± 0.5 |
| Acceleration Low (%) | Defender | 62.7 ± 17.3 | 60.2 ± 15.0 [a] | 63.4 ± 21.2 |
| | Blocker | 61.2 ± 16.9 | 61.7 ± 14.8 | 62.0 ± 17.4 |
| Acceleration Medium (%) | Defender | 21.5 ± 10.3 | 21.5 ± 7.8 | 20.0 ± 15.5 [b] |
| | Blocker | 23.6 ± 10.9 | 22.0 ± 9.1 [a] | 21.3 ± 13.2 |
| Acceleration High (%) | Defender | 15.6 ± 13.5 | 18.1 ± 12.8 [a] | 16.5 ± 12.9 |
| | Blocker | 15.1 ± 13.1 | 16.1 ± 11.9 | 16.5 ± 12.7 |
| Deceleration Low (%) | Defender | 58.4 ± 15.7 | 56.7 ± 14.1 [a] | 60.2 ± 15.5 |
| | Blocker | 60.6 ± 13.9 | 64.2 ± 13.0 [a] | 61.5 ± 14.0 [b] |
| Deceleration Medium (%) | Defender | 27.8 ± 12.9 | 28.0 ± 10.6 | 27.1 ± 14.0 |
| | Blocker | 25.9 ± 10.9 | 24.4 ± 8.7 | 26.1 ± 9.6 |
| Deceleration High (%) | Defender | 13.7 ± 13.0 | 15.2 ± 14.3 | 12.5 ± 11.3 |
| | Blocker | 13.4 ± 11.5 | 11.2 ± 10.8 [a] | 12.3 ± 12.1 |
| Jump Low (<20 cm) (%) | Defender | 19.6 ± 18.5 | 15.7 ± 12.0 [a] | 14.2 ± 16.3 [b] |
| | Blocker | 17.8 ± 15.5 | 15.0 ± 12.2 [a] | 18.6 ± 17.0 |
| Jump Medium (20–40 cm) (%) | Defender | 51.6 ± 22.0 | 53.5 ± 20.6 | 56.2 ± 26.0 |
| | Blocker | 47.5 ± 21.8 | 48.3 ± 22.1 | 46.0 ± 22.6 |
| Jump High (>40 cm) (%) | Defender | 28.6 ± 22.3 | 30.7 ± 20.5 | 29.4 ± 23.1 |
| | Blocker | 34.5 ± 25.2 | 36.6 ± 24.5 | 35.3 ± 22.6 |

[a] significantly different from Set 1 ($p < 0.05$); [b] significantly different from Set 2 ($p < 0.05$).

Regarding the distance covered at different intensities, Defenders increased their percentage of meters spent walking and decreased their jogging during the match, though significant differences were only observed between the second and the third set ($p = 0.005$ in walking, and $p = 0.004$ in jogging). Blockers increased their distance walking and decreased their distance jogging between the second and the third set ($p = 0.000$ in walking, and $p = 0.000$ in jogging).

Concerning acceleration variations during the match, Defenders decreased their percentage of acceleration at low intensity and increased this at high intensity between the first and the second set ($p = 0.013$ at low, and $p = 0.000$ at high intensity). Defenders also decreased the percentage of acceleration at medium intensity between the second and

the third set ($p$ = 0.012). Blockers decreased their percentage of acceleration at medium intensity between the first and the second set ($p$ = 0.003).

Defenders decreased their percentage of deceleration at low intensity between the first and the second set ($p$ = 0.048). Blockers increased the percentage of decelerated at low intensity ($p$ = 0.001) and decreased this at high intensity (0.001) between the first and the second set. Deceleration at low intensity decreased between the second and the third set ($p$ = 0.044).

Regarding jumps, only those performed with a lower elevation varied significantly during the match: Defenders decreased jumps' high from the first to second set ($p$ = 0.042), and also from second to the third set ($p$ = 0.004); Blockers also decreased elevation significantly between the first and the second set ($p$ = 0.049).

According to the match outcome (Table 4), the most important Defender characteristics that contributed to victory were high- and medium-intensity deceleration; distance jogging and quick running; acceleration (high, medium, and low); and Peak Player Load. Blocker characteristics that contributed to victory were maximum velocity and higher jumps, while acceleration (high and medium) and deceleration led to defeat.

**Table 4.** Discriminant function structure coefficients and tests of statistical significance of GPS variables that impact match outcome (victory and defeat) according to player position (Defender vs. Blocker) in female beach volleyball.

| Defenders Variables | SC | Blockers Variables | SC |
|---|---|---|---|
| Deceleration High | 0.55 * | Acceleration Higher | −0.66 * |
| Velocity Medium | 0.50 * | Deceleration Higher | −0.61 * |
| Velocity High | 0.49 * | Acceleration Medium | −0.51 * |
| Acceleration High | 0.48 * | Maximum Velocity | 0.49 * |
| Peak Player Load | 0.45 * | Jump Higher | 0.30 * |
| Acceleration Medium | 0.42 * | Velocity Higher | 0.22 |
| Acceleration Low | 0.37 * | Acceleration Lower | −0.22 |
| Deceleration Medium | 0.30 * | Jump Medium | 0.21 |
| Velocity Low | −0.26 | Meterage per Min | 0.17 |
| Deceleration Low | 0.25 | Jump Lower | 0.16 |
| Player Load | 0.18 | Deceleration Medium | −0.13 |
| Meterage per Min | −0.13 | Velocity Lower | 0.10 |
| Jump Medium | −0.05 | Distance Covered | 0.10 |
| Maximum Velocity | 0.03 | Deceleration Lower | −0.03 |
| Jump High | −0.01 | Velocity Medium | 0.02 |
| Distance Covered | 0.01 | Peak Player Load | 0.01 |
| Jump Low | 0.00 | Player Load | −0.01 |

* | SC | ≥ 0.30.

## 4. Discussion

With GPS technology, this research aimed to quantify the external workload of elite female BV players during competition matches with reference to player position and performance outcome. This technology is becoming quite popular in sports, although in BV, only a few studies have used this [9,16,17]. In fact, some of the variables used in other sports are not so useful to BV. For example, in most of the sports [14,20], total distance covered and Player Load express total work performed in a limited period of time. However, BV competition is not limited by time, but by points, which can run fast or slowly, thus influencing the total distance covered and Player Load in each set and match. Moreover, the third set finished at 15th point, so the displacement re-duce. Furthermore, the higher difference between team levels may dictate a lower distance covered if each rally could be shorter and sequenced [21]. The standard deviation observed in the distance covered in our study might reflect sets debated between more unbalanced teams [7]. Female BV is characterized by longer rallies than for male BV [22,23], which could highlight the total distance covered and Player Load. Addition-ally, variables that

depend on total distance to express different levels of intensity (displacement walking, jogging and running, acceleration, deceleration, and jumps) should not be used to compare players, sets or studies unless they are expressed by the percentage of the total distance, acceleration, deceleration, and fly-time (jumps).

Regarding the obtained results, differences between players' positions were found in most of the variables, being higher in Defenders (Peak Player Load, meters covered jogging and quick running, deceleration of medium- and high-intensity, and jump in medium-high zone) than in Blockers. These players only had higher values in meters covered in walking, and they had medium acceleration, low deceleration, and high jumps. These results may reflect the tactical system adopted by most BV teams. In this tactical system, Blockers try to make an impact through the block so they jump high, while Defenders have to perform several explosive movements that require multiple bouts of rapid accelerations and decelerations to avoid opponent points. This tactical strategy may be required to change as the opposing team may serve to the Defender, which force the Blocker to make the second touch, a pass. Overall, these results show that Defenders perform matches at higher intensity than Blockers, which is not consistent with the case study performed by Nunes et al. [9].

BV players changed their external load profile over two or three sets. As was expected, the distance covered and Player Load decreased significantly between the second and the third set because the last set was only points to 15, while the second set was points to 21, so the total absolute work was decreased by both Defenders and Blockers.

Besides this, Defenders displayed variations in Peak Player Load (a higher value in the second set), meterage per minute (decreased between the first and second set), maximal velocity (decreased between the first and second set), percentage of distance covered at different intensities (increased walking and decreased jogging), acceleration (decreased in low-intensity and increased in high-intensity between the first and second set), deceleration (decreased in low-intensity and increased in high-intensity be-tween the first and second set), and decreased jumps at low altitude. Altogether, these results suggest that in the second set, Defenders had to perform several intense acceleration and deceleration actions, which are very energy demanding and could account for a decrease in the output profile of GPS-related intense actions in the third set, suggesting the fatigue of the players. Fatigue has been defined as an ongoing dynamic process during high-intensity exercise that depends on central and peripheral mechanisms which limit the production of power efforts by the neuromuscular system [24]. Based on the reduction in the intensity od actions performed during the third set by Defenders, it is possible to assume that players experience increasing fatigue as the match progresses. It is very interesting to find that those mechanical actions that de-pend on the ability to accelerate (medium and high acceleration) and decelerate (medium and high deceleration) in a small space of action, including changing direction or rhythm (jogging and quick running) in response to an opponent's actions, reaching the ball, and generating opportunities to finish or create Blocker opportunities to success-fully finish the rally, were those discriminant variables that led Defenders to victory.

The output profile of Blockers over the sets revealed changes in meterage per minute (decreased between the first and second set), increased Peak Player Load, the percentage of meters covered walking (increased between the first and the second set) and jogging (decreased between the first and the second set), decreased acceleration and deceleration intensity, and a decreased percentage of lower jumps. In general, these results evidenced the higher offensive action of Blockers, probably during the at-tack and spikes once the frequency of jumps performed at higher and medium altitude increased. While Defenders exhibited a profile consistent with fatigue, Blockers' variation along the sets was not constant, and did not suggest fatigue. The intensity of acceleration performed by Blockers increased along the match, which could reflect the necessity to compensate for some of the fatigue of Defenders, substituting them in movements such as acceleration to receive or pass, or this may suggest changes in the opposing team's strategy in increasing services

directed to Blockers. Although no significant differences were found in the years of BV-playing experience, Blockers had on average of almost three years more experience than Defenders, which may contribute to improve the relationship between periods of intense effort and recovery [16].

The main actions performed by Blockers were jumps, either to block or to spike. The Blocker's biggest displacement actions included movement from the service area to the net, and along the net [9,17]. These actions were consistent with the higher distance walking in all sets than in other intensities. Our results support the idea that medium and high acceleration, and high deceleration, are the Blocker variables that most likely lead to defeat, while maximal velocity and high jumps contribute to victory. These results enhance the importance of jumping high in the block and spike actions, decisive both to avoid opponent points and to score. Moreover, acceleration and deceleration are important components of quick, short, and intermittent movements more related to Defenders. Despite these novel findings regarding female BV players, the authors acknowledge that the current research is a pilot study with only 12 players, which makes it difficult to discuss this study and compare it to others. Additionally, only one championship has been analyzed, related to a high-performance level in the senior women's category.

Different methodologies have been used to characterize external BV, but it seems that future research should include information on training load (internal and external) and match analysis.

## 5. Conclusions

Our results support the idea that medium and high acceleration and high deceleration are the Blocker variables that most likely lead to defeat, while maximal velocity and high jumps contribute to victory. These results enhance the importance of jumping high in the block and spike actions, which is decisive both to avoid opponents scoring points and for scoring. To improve jumping performance, strength gain is an important component that should be considered in training periodization. Training load should be progressive (frequency, volume, and intensity), starting during the preparation phase with general strength methods that precede maximal strength and power training in subsequent stages to increase jumping ability.

Acceleration and deceleration are important components of quick, short, and intermittent movements more related to Defenders. Intense acceleration and deceleration are very intense actions that increase the risk of injury. To avoid these situations, athletes should be exposed to strength training and high speeds during preseason. Exercises that increase stiffness are essential to improve powerful hip motion, and to reduce energy leaks in the trunk. Some examples of these exercises are jerks, cleans, weighted step-us, squats, hack backs, and single-leg squats.

Female BV training should focus on different training methodologies according to a player's position. For Blockers, the training program should prioritize exercises that enhance maximum velocity and the ability to perform high-intensity jumps, followed by less intense periods to facilitate recovery. The Defenders' program should incorporate an instability routine and explosive exercises to balance modified ground reaction forces in sand, clearly depending on high acceleration, deceleration, and velocity actions. This should also be followed by less intense periods to facilitate recovery.

**Author Contributions:** Conceptualization, P.V.J. and M.P.M.; methodology, P.V.J., A.M., and M.P.M.; software, M.L. and H.O.; validation, M.L., H.O., and P.V.J.; formal analysis, P.V.J. and M.P.M.; investigation, P.V.J. and M.P.M.; resources, P.V.J., M.L. and H.O.; data curation, M.P.M.; writing—original draft preparation, P.V.J. and M.P.M.; writing—review and editing, P.V.J., M.P.M. and H.O.; visualization, M.L.; supervision, P.V.J., M.P.M., A.M.; project administration, P.V.J.; funding acquisition, P.V.J., EO and M.L. All authors have read and agreed to the published version of the manuscript.

**Funding:** This research had no funding, but a loan from Catapult, as the research team did not have funding for the purchase of the assessment instruments.

**Institutional Review Board Statement:** Written informed consent was obtained from players before the investigation according to the Declaration of Helsinki and the University of Trás-os-Montes and Alto Douro Ethics Committee approved the study.

**Informed Consent Statement:** Informed consent was obtained from all subjects involved in the study.

**Acknowledgments:** The authors would like to thank all coaches, staff members, and players used in this beach volleyball study for their active participation and to Catapult for lending the instruments used for carrying out the research.

**Conflicts of Interest:** The authors declare no conflict of interest.

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
