# Peer review of "Global Position Analysis during Official Elite Female Beach Volleyball Competition: A Pilot Study"

_applsci, doi:10.3390/app11209382_

Round 1
Reviewer 1 Report
- Maybe the authors should consider using a Binomial test especially for testing the percentage variables instead of using only the Mann-Whitney U test and Wilcoxon test.
- Line 130 inside the table 2. add the word ''Variables'' for all tested data of the study in the upper left corner of the table and add (*) in p values for the variable ''Deceleration Low (%)
- Lines 186-190 are not appropriate for the discussion section. The authors can move these lines to the introduction section and rewrite the section accordingly.
- Lines 229-232 are not appropriate for the discussion section. The authors can move these lines to the introduction section and rewrite the section accordingly.
- Add strong and weak points e.g. limitations of your study, for example the use of only 12 players (despite the truth that they were the top 12 players of the Portuguese national Beach Volleyball Championship) and that the results should use with cautions
Author Response
We hereby thank you for your comments and suggestions. In this sense, the changes made are described below, in the hope that with these changes made, readers can become more enlightened in relation to our study objective and contribution to the practical knowledge of this research.
A binomial test uses sample data to determine if the population proportion of one level in a binary (or dichotomous) variable equals a specific claimed value. In this study the variables analyzed are not dichotomous and don’t have an expected specific value to make comparison.
- Done
- Inserted in line 63
- Lines 229-232 are discussion our results and are not describing results from other studies.
- We include the following sentence in line 245: “Despite these novel findings with female BV players, the authors acknowledge that the current research is a pilot study with only 12 players that made it difficult to discuss this study and compare it to others. Additionally, only one championship has been analyzed, related to a high-performance level in the senior women’s“
Reviewer 2 Report
Thank you for your very well-written submission. It was a delight to read.
A paper’s purpose statement is so crucial. Consider adding a clear and detailed purpose statement early in the manuscript – before the method section.
An N= 12 is too small for results to be validated or powerful. That is not to say the findings are uninteresting or unworthy of further investigation. That said, this study needs to be termed something other than a study. Perhaps a “pilot study” or “exploratory investigation”. But this research cannot be validated with an N= of 12. I think it can still fit in this journal, as it is in scope, and generally interesting, if it’s re-termed an “exploratory” or “pilot study”.
That said, the writing is good, the flow makes conceptual sense, and clarity is demonstrated throughout. It follows the sequence of research studies in this line.
A stronger rationale for this study needs to be expressed. It is interesting on its own face value. But connecting this line of inquiry to results which have impact on sport performance, training methodology, exercise physiological applications is required to give this line of inquiry more meaning to a broader readership. I see an effort to do this was made in the first paragraph of the discussion section. I recommend a stronger devotion to that effort. Perhaps in the conclusion the author could draw a connection to scholarship which has efforted to connect training to injury prevention to support a stronger rationale to your study. I believe the conclusion of your article could emphasize a connection to training to better rationalize your research efforts. The citation for that work is below:
Pennington, C. G. (2020). Strength and Conditioning in Women’s College Volleyball: Anaerobic Power Increases and Injury Prevention. Journal of Physical Fitness, Medicine & Treatment in Sports. 8(4), 31-31. DOI: 10.19080/JPFMTS.2020.08.555742.
Thank you for this contribution. It is well done and I believe it worthy of publication once these requests are made:
- Consider adding a purpose statement early in the manuscript – before the method section.
- Re-term your work pilot study or exploratory study.
- Develop a stronger rationale to the study (perhaps emphasizing the importance of training for volleyball athletes).
- Consider using the source provided to assist in the rationale for your work.
Author Response
We hereby thank you for your comments and suggestions. In this sense, the changes made are described below, in the hope that with these changes made, readers can become more enlightened in relation to our study objective and contribution to the practical knowledge of this research.
-A detailed purpose was added: “So, the aim of this pilot study was to quantify through the use of GPS technology external workload (player load, distance covered, acceleration/deceleration, maximal speed and jumps) able to predict the final scores in Women BV Portuguese Championship according to player position (blocker and defender). The specific aims of the study were: a) to characterize athletes’ profile by position based on external workload variables, b) to analyze external workload variables along the three sets with reference to the player position, and c) to analyze those external workload variables that discriminate to victory or defeat for each player position.“
- The title was changes for “Global Position Analysis during Official Elite Female Beach Volleyball Competition: A pilot study“
Included in line 77 :” Physical demands monitoring can provide important information that allow staff to use more effective training prescription to maximize athletes’ performance, encourage recovery and prevent injury.”
- The conclusion was modified in order to include more information about training preparation according to players’ position.
- Our results support that medium and high acceleration and high deceleration are the Blockers variables that most discriminate to defeat while maximal velocity and high jumps discriminate to victory. These results enhance the importance of jumping high in the block and spike actions, decisive both to avoid opponents point and to score. To improve jumping performance, strength gain is an important component that should be considered in training periodization. Training load should be progressive (frequency, volume and intensity) starting during the preparation phase with general strength methods that precede maximal strength and power training in subsequent stages to increase jumping ability.
- Acceleration and deceleration are important components of quick, short and intermit-tent movements more related to Defenders. Intense accelerations and deceleration are very intense actions that increase the risk of injury. To avoid these situations athletes should be exposed to strength training and high speed during the preseason. Exercises that increase stiffness, improve powerful hip motion and reduce energy leaks in the trunk are essential. Some examples of these exercises are: jerks, cleans, weighted step-us, squats, hack backs and single-leg squats.
- Female BV training should focus on different training methodologies according to players’ position. Blockers’ training program should prioritize exercises that enhance maximum velocity and the ability to perform high-intensity jumps followed by less in-tense periods to facilitate recovery. The Defender's specific program should incorporate instability routine and explosive exercises to balance modified ground reaction forces in sand, clearly depending on high acceleration, deceleration and velocity actions, that should also be followed by less intense periods to facilitate recovery.
Round 2
Reviewer 2 Report
The authors did a fine job addressing my most pressing concerns. I believe the literature review and conclusions could include more relevant citations to connect the rationale of the present pilot study to work done in the past.